# FULLY HYPERBOLIC REPRESENTATION LEARNING ON KNOWLEDGE HYPERGRAPH

## ABSTRACT

Knowledge hypergraphs generalize knowledge graphs in terms of utilizing hyperedges to connect multiple entities and represent complicated relations within them. Existing methods either transform hyperedges into an easier to handle set of binary relations or view hyperedges as isolated and ignore their adjacencies. Both approaches have information loss and may lead to sub-optimal models. To fix these issues, we propose the Hyperbolic Hypergraph GNN ($H^2$GNN), whose essential part is the *hyper-star message passing*, a novel scheme motivated by a lossless expansion of hyperedges into hierarchies, and implement a direct embedding which explicitly takes adjacent hyperedges and entity positions into account. As the name suggests, $H^2$GNN works in the fully hyperbolic space, which can further reduce distortion and boost efficiency. We compare $H^2$GNN with 15 baselines on both homogeneous and heterogeneous knowledge hypergraphs, and it outperforms state-of-the-art approaches in both node classification and link prediction tasks.

## 1 INTRODUCTION

Knowledge hypergraphs are natural and straightforward extensions of knowledge graphs (Chen et al., 2023; Wang et al., 2023a). They encode high-order relations within diverse entities via hyper-relations and have been widely used in downstream tasks including question answering (Jia et al., 2023; Guo et al., 2021), recommendation system (Yu et al., 2021; Tan et al., 2011), computer vision (Li et al., 2023; Zeng et al., 2023) and healthcare (Wu et al., 2023). Generally, knowledge hypergraphs store factual knowledge as tuples (relation, entity$_1$, ..., entity$_m$), where entities correspond to nodes and hyper-relations correspond to hyperedges.

The core of representation learning for knowledge hypergraphs lies in the embedding of hyperedges. Existing methods can be roughly classified into two groups. One is the indirect way (Guan et al., 2020a; Fatemi et al., 2020a; Rosso et al., 2020a), i.e., they transform hyperedges into a set of binary relations and then apply the methods of knowledge graphs. However, the transformation is lossy and may hinder the performance. Take '(flight, Beijing, Shanghai, Guangzhou)' as an example, which means the flight takes off from Beijing and passes through Shanghai before landing in Guangzhou. Accordingly, it will be split into three distinct triples: '(Beijing, flight, Shanghai)', '(Shanghai, flight, Guangzhou)', and '(Beijing, flight, Guangzhou)', which lose the crucial information that Shanghai is an intermediate location and introduces unreal flight from Beijing to Guangzhou. This example also discloses that the order of entities is highly related to the semantics, i.e., the entity position in hyperedges is important.

The other is the direct way (Wen et al., 2016; Fatemi et al., 2020a; Guan et al., 2019). However, these methods commonly view hyperedges as isolated and learn embeddings independently, which may lose information essential for the downstream tasks. For example, consider the tuples '(education, Stephen Hawking, University College Oxford, BA degree)'. Obviously '(locate, University College Oxford, Oxford, England)' is the adjacent hyperedge since they have node 'University College Oxford' in common. By putting together, it can be inferred that the entity 'Stephen Hawking' is 'person' and the hyper-relation 'live in' in the knowledge hypergraph should also include '(live in, Stephen Hawking, 1959-1962, England)'. Therefore, appropriately incorporating adjacencies is crucial.

With these observations, we consider equipping Graph Neural Networks (GNN) with a hypergraph-specialized hyper-star message passing scheme, drawing inspiration from a lossless hyper-star expansion. Specifically, we introduce position-aware representations for each node, and then, in

each GNN layer, a two-stage message passing is performed, one is the aggregation of hyperedges embedding through the nodes they contain, while the other focuses on updating each node embedding by considering their positions and adjacent hyperedge embeddings. Furthermore, the hierarchy demonstrated in the message passing process inspires us to explore a representation in a fully hyperbolic space that can better capture the characteristics of scale-free and hierarchical graphs (Shi et al., 2023; Chami et al., 2019; Krioukov et al., 2010; Muscoloni et al., 2017; Chen et al., 2022). We notice that Fan et al. (2021) also utilizes GNN for learning knowledge hypergraph, however, it models hyperedge in a class-dependent way, that is for multiple type hyper-relations, they need to create a separate hypergraph for each type of hyper-relation, which cannot be satisfied in most situations. Instead, we view hyperedges as instance-dependent, modeling them based on actual instances directly with a variety of hyperedge types and node types, and propose the corresponding hyper-star message passing scheme. The contributions of the paper are threefold:

- We propose a novel hypergraph-specific message passing scheme, which can be seamlessly integrated into any mainstream GNN.

- We make the first attempt to apply GNN for modeling hyper-relations in an instance-dependent way.

- We implement a versatile plug-and-play encoder, which can be easily concatenated with task-specific decoders and widely used in a wide range of downstream tasks.[1]

## 2 PRELIMINARIES

In this section, we present the notation used and provide an overview of the prior knowledge utilized in the proposed method.

**Representation Learning Problem Definition.** In knowledge hypergraphs, each tuple $(r, x_1, x_2, ..., x_m)$ represents a knowledge fact, where $x_1, x_2, ..., x_m$ denote the entities, and $r$ represents the hyper-relation, where $m$ is called the arity of hyper-relation $r$. We convert the knowledge tuples to hypergraph $\mathcal{G} = (\mathcal{V}, \mathcal{R}, \mathcal{E})$, where $\mathcal{V}$ denotes the set of entities, $\mathcal{E}$ is the set of hyperedges, $\mathcal{R}$ denotes the set of hyper-relations. The goal of representation learning is to obtain embedded representations for each node $x \in \mathcal{V}$ and relation type $r \in \mathcal{R}$ in the knowledge hypergraphs.

**Message Passing.** General GNNs leverage both the feature matrix and graph structure to obtain informative embeddings for a given graph. The node embeddings undergo iterative updates by incorporating information from their neighboring nodes. The message-passing process in the $l$-th layer of GNN is formulated as follows:

$$\boldsymbol{x}_i^{l+1} = \phi^l(\boldsymbol{x}_i^l, \{\boldsymbol{x}_j^l\}_{j \in \mathcal{N}_i}),$$

where $\mathcal{N}_i$ denotes the collection of neighboring nodes of node $x_i$. $\phi^l$ defines the aggregation operation of the $l$-th layer. Message passing process in homogeneous hypergraphs is summarized as follows:

$$\begin{cases} \boldsymbol{h}_e = \phi_1(\{\boldsymbol{x}_j\}_{j \in e}), \\ \boldsymbol{x}_i = \phi_2(\boldsymbol{x}_i, \{\boldsymbol{h}_e\}_{e \in \mathcal{E}_i}), \end{cases}$$

where $\mathcal{E}_i$ represents the set of all hyperedges that contain node $x_i$, the given equation utilizes two permutation-invariant functions $\phi_1$ and $\phi_2$ to aggregate messages from nodes and hyperedges respectively (Huang & Yang, 2021).

**Hyper-star Expansion.** Specifically, for the hypergraph $\mathcal{G} = (\mathcal{V}, \mathcal{R}, \mathcal{E})$, We expand it into a new heterogeneous graph $\mathcal{G}^* = (\mathcal{V}^*, \mathcal{R}^*, \mathcal{E}^*)$ by introducing a new node for each hyperedge $e \in \mathcal{E}$ and connecting it to the nodes contained in this hyperedge. Thus, $\mathcal{G}^*$ includes both the original nodes from $G$ and generated nodes transformed from hyperedges in $\mathcal{E}$. Newly generated nodes are connected with other nodes in the graph based on different types of hyperedges and positions of nodes within them, i.e. $\mathcal{E}^* = \{(r^*, u, e) : r^* = \mathcal{T}(e)\_i, u \in \mathcal{V}, e \in \mathcal{E}\}$, where the function $\mathcal{T}$ maps hyperedges to their respective relations and $i$ represents the position of node $u$ within hyperedge $e$.

---

[1]The source code will be released after the paper is accepted.

As shown in Figure 1, the hyperedges *TeamRoster* and *SportAward* are transformed into a new node and connected with the nodes previously contained in the hyperedge. The newly generated relation is determined by the type of the hyperedge and the position of the node in the hyperedge.

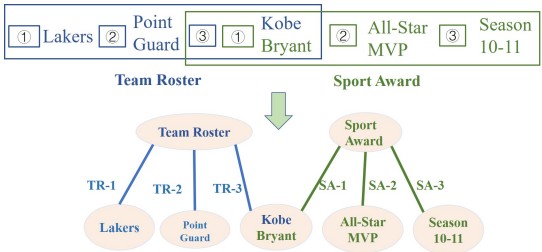

Figure 1: The hyper-star expansion process. We introduce position-aware relation embedding such as 'TR-1' 'SA-3', etc., where 'TR' 'SA' are abbreviations of relation 'Team Roster' and 'Sport Award', the numbers '1' '2' '3' denote the position.

**Hyperbolic Space Explanation.** Previous studies have utilized various hyperbolic geometric models, including the Poincaré ball model (Ungar, 2001), the Poincaré half-plane model (Stahl, 1993), the Klein model (Visser, 1985), and the hyperboloid (Lorentz) model (Bobylev et al., 1997). In Figure 2, when we embed the tree structure into Euclidean space, the distance between the yellow and pink nodes on the tree is 8 nodes apart, but in reality, these tree-like structures are very close in real networks (Kennedy et al., 2013; Adcock et al., 2013). Furthermore, in Euclidean space, volume growth occurs at a polynomial rate, as seen in the left of the figure where the network space expands quadratically with radius. In contrast, hyperbolic space exhibits exponential volume growth, mirroring the tree structure observed in real networks. Consequently, distortions occur when Euclidean space lacks the capacity to accommodate a multitude of nodes, emphasizing the need to consider the incorporation of hyperbolic spaces.

We denote a hyperboloid model $\mathbb{H}_k^n$ with negative curvature $k$ in $n$ dimensions. The tangent space at $\boldsymbol{x}$ in $\mathbb{H}_k^n$ is an $n$-dimensional vector space that approximates $\mathbb{H}_k^n$:

$$\Gamma_x \mathbb{H}_k^n := \{\boldsymbol{v} \in \mathbb{R}^{n+1} : \langle \boldsymbol{v}, \boldsymbol{x} \rangle_{\mathbb{H}} = 0\},$$

Where $\langle \boldsymbol{v}, \boldsymbol{x} \rangle_{\mathbb{H}}$ is the hyperboloid inner product $\langle \boldsymbol{v}, \boldsymbol{x} \rangle_{\mathbb{H}} = \boldsymbol{v}^T diag(-1, 1, ...1, 1)\boldsymbol{x}$. $\Gamma_x$ defines the mapping from hyperboloid space to tangent space. The mapping relation between the manifold $\mathbb{H}_k^n$ and its tangent space $\Gamma_x \mathbb{H}_k^n$ can be established using the exponential and logarithmic map.

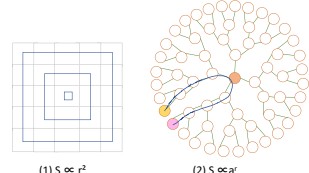

Figure 2: Comparison of Euclidean and Hyperbolic Spaces.

# 3 PROPOSED METHOD

In this section, we describe H²GNN in detail for linear transformation and hyper-star message passing. The overall architecture is shown in Figure 3.

## 3.1 LINEAR TRANSFORMATION

We first implement a matrix function to perform linear transformations in hyperbolic space, laying the foundation for the development of hyperbolic graph neural networks. Follow the (Chen et al., 2022), we transform the linear layer problem in hyperbolic space into a learning process of matrix $\mathbf{M} = [\boldsymbol{v}^\top; \mathbf{W}]$, where $\boldsymbol{v} \in \mathbb{R}^{n+1}$, $\mathbf{W} \in \mathbb{R}^{m \times (n+1)}$. This matrix should satisfy the condition that for all $\boldsymbol{x} \in \mathbb{H}^n$, $\mathcal{F}_x(\mathbf{M})\boldsymbol{x} \in \mathbb{H}^m$, where $\mathcal{F}_x : \mathbb{R}^{(m+1) \times (n+1)} \to \mathbb{R}^{(m+1) \times (n+1)}$ transforms any matrix into an appropriate value that minimizes the loss function. The implementation of the fully Hyperboloid linear layer is as follows:

$$y = HL(\boldsymbol{x}) = \left[ \sqrt{\|\phi(\mathbf{W}\boldsymbol{x}, \boldsymbol{v})\|^2 - 1/k}; \phi(\mathbf{W}\boldsymbol{x}, \boldsymbol{v}) \right]^\top, \quad (1)$$

where $\boldsymbol{x} \in \mathbb{H}_k^n$, $\mathbf{W} \in \mathbb{R}^{m \times (n+1)}$, and $\phi$ is an operation function. For dropout, the function can be expressed as $\phi(\mathbf{W}\boldsymbol{x}, \boldsymbol{v}) = dropout(\mathbf{W}\boldsymbol{x})$. For activation and normalization, the function can be written as:

$$\phi(\mathbf{W}\boldsymbol{x}, \boldsymbol{v}) = \frac{\lambda \sigma(\boldsymbol{v}^\top \boldsymbol{x} + b')}{\|\mathbf{W}h(\boldsymbol{x}) + \boldsymbol{b}\|}(\mathbf{W}h(\boldsymbol{x}) + \boldsymbol{b}) \quad (2)$$

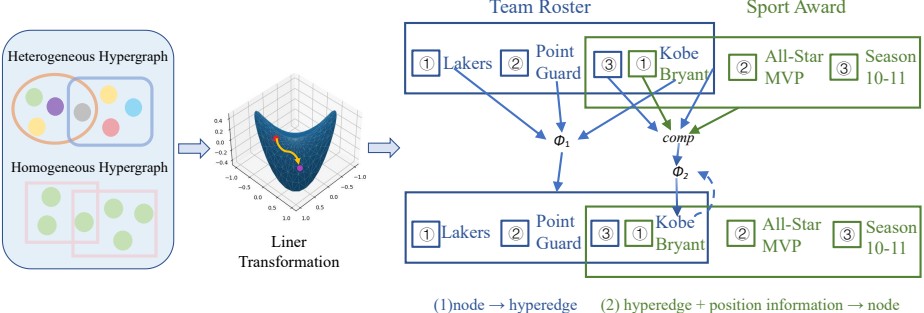

Figure 3: Overall of H$^2$GNN. Blue represents the TeamRoster hyper-relation and its contained nodes, while green represents the Sport Award hyper-relation and nodes. We display the aggregation and composition operations in message passing. The blue arrows indicate the process of updating $TeamRoster$ hyperedge embedding through $\phi_1$ aggregation operation. The green arrows depict the update of $KobeBryant$ via composition operation. The dashed line represents the message passing of the node embedding of $KobeBryant$ before updating the node.

where $\sigma$ represents the sigmoid function; $\boldsymbol{b}$ and $b'$ are biases; $\lambda > 0$ controls scaling range; and $h$ denotes the activation function. Linear transformation guarantees the outputs remain in the hyperbolic space, a detailed proof and derivation process can be found in (Chen et al., 2022).

## 3.2 Hyper-star Message Passing

The existing message passing framework, as discussed in Section 2, is insufficient for handling the challenges posed by knowledge hypergraphs, requiring consideration of two additional aspects. 1) Knowledge hypergraphs frequently encompass diverse relation types, necessitating the integration of different relation type information into the message-passing process. 2) Hyperedges are typically represented by tuples (i.e., $(r, e_1, e_2...e_m)$), where the order of entities in an $m$-tuple indicates their role in the relation, akin to subject and object roles in simple graphs. Therefore, we contend that position information regarding entities participating in relations must be considered during the message-passing process.

We introduce a $d$-dimensional position-aware fearure $\boldsymbol{h}_p \in \mathbb{R}^d$, which is relation-type specialized. That means that hyperedges of the same relation type have the same relation-position representations. To incorporate position embedding into the message passing, we leverage the composition operations. The equation can be written as:

$$\begin{cases} \boldsymbol{h}_e = \phi_1(\{\boldsymbol{x}_j\}_{j \in e}) \\ \boldsymbol{x}_i = \phi_2(\boldsymbol{x}_i, comp(\boldsymbol{h}_e, \boldsymbol{h}_p)_{e \in \mathcal{E}_i}), \end{cases} \quad (3)$$

where $\boldsymbol{h}_p$ varies with different relation types and different positions within the same relation type, implying both relation type and position information. $comp$ represents a composition operation that is utilized to integrate $\boldsymbol{h}_p$ into the process of information transmission.

In the first stage, we utilize $\phi_1$ to aggregate features of all nodes within each hyperedge $e$. In the second stage, we use $\phi_2$ to update the embedding of each node based on the associated hyperedge, where position-aware embeddings are incorporated into the message passing process through composition operation $comp$. We evaluate one non-parametric simple operator in hyperbolic space, defined as: $comp(\boldsymbol{h}_e, \boldsymbol{h}_p) = \boldsymbol{h}_e - \boldsymbol{h}_p$. Moreover, $\phi_1$ and $\phi_2$ are implemented through additive aggregation operations, ensuring that these operations remain within the hyperbolic space. Section 4 demonstrates the effective performance of our encoder based on the simple design. In summary, we present a straightforward yet highly effective instantiation framework, which considers the valuable neighborhood information:

$$\begin{cases} \boldsymbol{h}_e = centroid(\{\boldsymbol{x}_j\}_{j \in e}) \\ \boldsymbol{x}_i = centroid(\boldsymbol{x}_i, (\boldsymbol{h}_e - \boldsymbol{h}_p)_{e \in \mathcal{E}_i}), \end{cases} \quad (4)$$

where $centroid$, as demonstrated in (Law et al., 2019), is used to determine the center point of hyperboloid space:

$$centroid\{\boldsymbol{x}_j\}_{j \in e} = \frac{\sum_{j \in e} \boldsymbol{x}_j}{\sqrt{-K} |\|\sum_{j \in e} \boldsymbol{x}_j\|_{\mathbb{H}}|} \tag{5}$$

where $K$ is a negative curvature, $\|a\|_{\mathbb{H}}^2 = \langle a, a \rangle_{\mathbb{H}}$ is the squared Lorentzian norm of $a$.

## 3.3 Objective Function and Training

For the node classification task, we use the negative log-likelihood loss to optimize our model by minimizing the difference between the predicted log probabilities and the ground truth labels of each node.

$$\mathcal{L} = -\sum_{i=1}^{N} \sum_{j=1}^{C} 1_{\{y_i=j\}} \log P(y_i = j | \mathbf{x}_i) \tag{6}$$

where $N$ represents the number of entities, $C$ represents the number of labels. The indicator function $1_{\{y_i=j\}}$ takes the value of 1 when the gold label $y_i$ equals $j$, and 0 otherwise. $P(y_i = j | \mathbf{x}_i)$ represents the model's predicted probability that node $i$ belongs to label $j$.

For the link prediction task, we train our model on both positive and negative instances, which are generated using the same method as HypE (Fatemi et al., 2020b). Specifically, we create $N * r$ negative samples for every positive sample from the dataset by randomly replacing each correct entity with $N$ other entities. Here, $N$ is a hyperparameter and $r$ is the number of entities in the tuple. The dataset is divided into three subsets: the training set $\mathcal{E}_{train}$, the test set $\mathcal{E}_{test}$, and the validation set $\mathcal{E}_{valid}$. These sets contain the correct tuples for each category, and $\mathcal{E} = \mathcal{E}_{train} \cup \mathcal{E}_{test} \cup \mathcal{E}_{valid}$. For any tuple $\boldsymbol{x} \in \mathcal{E}$, $neg(\boldsymbol{x})$ is utilized to generate a set of negative samples, following the aforementioned process. To compute our loss function, we define the cross-entropy loss as follows:

$$\mathcal{L} = \sum_{\boldsymbol{x} \in \mathcal{E}_{train}} - \log \frac{\exp g(\boldsymbol{x})}{\exp g(\boldsymbol{x}) + \sum_{\boldsymbol{x}' \in neg(\boldsymbol{x})} \exp g(\boldsymbol{x}')} \tag{7}$$

where $g(\boldsymbol{x})$ predicts the confidence score of the tuple $\boldsymbol{x}$.

## 4 Experiments

In this section, we evaluate H$^2$GNN in transductive learning tasks, specifically node classification and link prediction. Similar to (Huang & Yang, 2021), we conduct inductive learning tasks for homogeneous hypergraphs in the Appendix A.1. Given a hypergraph $\mathcal{G}$, consisting of node data $\mathcal{V}$ and hyperedges $\mathcal{E}$, the node classification and inductive learning tasks involve developing a classification function that assigns labels to nodes. The link prediction task focuses on predicting new links between entities within the hypergraph, leveraging the existing connections as a basis.

### 4.1 Settings

**Dataset.** We employ widely used academic Co-citation and Co-author datasets (Yadati et al., 2019), including DBLP, CiteSeer, Pubmed, and Cora for node classification tasks. For the link prediction task, our approach is evaluated on two hyper-relation datasets: JF17k (Wen et al., 2016) and FB-AUTO (Bollacker et al., 2008), which consist of both binary and $n$-ary facts. Further details and statistics of the datasets can be found in Table 1.

**Compared methods.** For the node classification task, we conduct a comparative analysis between H$^2$GNN and representative baseline methods, including Hypergraph neural networks (Feng et al., 2019), HyperGCN (Yadati et al., 2019), FastHyperGCN (Yadati et al., 2019), HyperSAGE (Arya et al., 2020), UniGNN (Huang & Yang, 2021). In the knowledge hypergraph link prediction, we categorize the introduced baselines into two groups: (1) models that operate with binary relations and can be easily extended to higher-arity: r-SimplE (Fatemi et al., 2020a), m-DistMult (Fatemi et al., 2020b), m-CP (Fatemi et al., 2020b) and m-TransH (Wen et al., 2016); and (2) existing methods capable of handling higher-arity relations: NeuInfer (Guan et al., 2020b), HINGE (Rosso et al., 2020b), NaLP (Guan et al., 2019), RAE (Zhang et al., 2018) and HypE (Fatemi et al., 2020b).

Table 1: Statistics on the dataset, 'classes' and 'relations' are the number of node types and hyperedge types, respectively.

| | DBLP (Co-authorship) | Cora (Co-authorship) | Cora (Co-citation) | Pubmed (Co-citation) | Citeseer (Co-citation) | JF17K (Knowledge Base) | FB-AUTO (Knowledge Base) |
|---|---|---|---|---|---|---|---|
| hypernodes | 43,413 | 2,708 | 2,708 | 19,717 | 3,312 | 29,177 | 3,388 |
| hyperedges | 22,535 | 1,072 | 1,579 | 7,963 | 1,079 | 102,648 | 11,213 |
| classes | 6 | 7 | 7 | 3 | 6 | - | - |
| relations | - | - | - | - | - | 327 | 8 |
| #2-ary | 9,976 | 486 | 623 | 3,522 | 541 | 56,332 | 3,786 |
| #3-ary | 4,339 | 205 | 464 | 1,626 | 254 | 34,550 | 0 |
| #4-ary | 2,312 | 106 | 312 | 845 | 118 | 9,509 | 215 |
| #5-ary | 1,419 | 78 | 180 | 534 | 65 | 2,230 | 7,212 |
| #6-ary | 906 | 45 | 0 | 297 | 40 | 37 | 0 |

**Hyper-parameter setting.** We implemented the $H^2$GNN framework using PyTorch and performed the training process on a Tesla V100 GPU machine. The parameters for the other methods are configured according to the recommendations provided by their respective authors. For the node classification task, we adopt a two-layer $H^2$GNN with the following hyper-parameters: learning rate of $0.01$, weight decay of $5e-5$, the dropout rate of $0.5$, and hidden layer dimension of $8$. We fix the number of training epochs at $200$ and report model performance based on the best validation score on the test dataset for each run. For the link prediction task, we adopt a single-layer $H^2$GNN with the following hyper-parameters: learning rate of $0.05$, embedding dimension of $200$, the dropout rate of $0.2$, and a negative ratio of $10$. We trained using batches of $128$ items for $2000$ iterations, selecting the model that achieved the highest validation score for testing purposes and recording its results.

## 4.2 NODE CLASSIFICATION RESULTS

Table 2: The accuracy(%) of node classification on co-authorship and co-citation datasets for baseline methods and $H^2$GNN. The best and most competitive results are highlighted for each dataset.

| Method | Co-authorship Data | | Co-citation Data | | |
|---|---|---|---|---|---|
| | DBLP | Cora | Cora | Pubmed | Citeseer |
| UniSAGE | 88.29±0.22 | 74.04±1.50 | 67.08±2.32 | 74.34±1.56 | 61.27±1.78 |
| UniGIN | 88.34±0.21 | 73.82±1.36 | 66.94±2.07 | 74.46±1.81 | 61.09±1.60 |
| HyperSAGE | 77.25±3.11 | 72.21±1.40 | 66.84±2.27 | 72.33±1.18 | 61.08±1.72 |
| HyperGCN | 71.17±8.73 | 63.29±7.11 | 62.43±9.17 | 67.91±9.43 | 57.98±7.01 |
| FastHyperGCN | 67.86±9.46 | 61.60±7.99 | 61.42±10.03 | 65.17±10.03 | 56.76±8.10 |
| HGNN | 68.08+5.10 | 63.21±3.02 | 68.01±1.89 | 66.45±3.17 | 56.99±3.43 |
| **$H^2$GNN (Ours)** | **89.75±0.20** | **74.97±1.20** | **69.43±1.54** | **74.89±1.23** | **62.52±1.48** |

Table 2 presents the node classification accuracy, We observe that $H^2$GNN significantly outperforms other methods on all datasets, achieving an accuracy range of 89.75% to 74.89%, with a low standard deviation of 0.20% to 1.54%. This demonstrates that $H^2$GNN can effectively capture the structure information of the hypergraph, thereby improving the performance and stability of the node classification task. Furthermore, when compared to UniSAGE, which also employs a two-stage message passing schema in the Euclidean space, it becomes evident that hyperbolic space is better suited for modeling hierarchical structural information.

## 4.3 KNOWLEDGE HYPERGRAOH LINK PREDUCTION

*Knowledge hypergraph completion* can be achieved by either extracting new facts from external sources or predicting links between existing facts in the hypergraph. The latter entails inferring new knowledge from the structure of the hypergraph itself, which is the focus of our experiment (Rossi et al., 2021). Table 3 presents the results on two datasets across relational knowledge bases. We employ $H^2$GNN as the encoder and m-DistMult as the decoder, achieving the highest values on Hits@10 evaluation metrics, with scores of $0.869$ on FB-AUTO and $0.660$ on the JF17k dataset. Our method demonstrates a significant improvement over G-MPNN, which was specifically designed for

Table 3: Knowledge Hypergraph link prediction results on JF17k and FB-AUTO for baselines and H$^2$GNN. The G-MPNN method did not produce results on the JF17k dataset for two days, so the experimental results are not shown.

| Method | FB-AUTO | | | | JF17K | | | |
|---|---|---|---|---|---|---|---|---|
| | Hits@1 | Hits@3 | Hits@10 | MRR | Hits@1 | Hits@3 | Hits@10 | MRR |
| m-TransH | 0.602 | 0.754 | 0.806 | 0.688 | 0.370 | 0.475 | 0.581 | 0.444 |
| m-CP | 0.484 | 0.703 | 0.816 | 0.603 | 0.298 | 0.443 | 0.563 | 0.391 |
| m-DistMult | 0.513 | 0.733 | 0.827 | 0.634 | 0.372 | 0.510 | 0.634 | 0.463 |
| r-SimplE | 0.082 | 0.115 | 0.147 | 0.106 | 0.069 | 0.112 | 0.168 | 0.102 |
| NeuInfer | **0.700** | 0.755 | 0.805 | 0.737 | 0.373 | 0.484 | 0.604 | 0.451 |
| HINGE | 0.630 | 0.706 | 0.765 | 0.678 | 0.397 | 0.490 | 0.618 | 0.473 |
| NALP | 0.611 | 0.712 | 0.774 | 0.672 | 0.239 | 0.334 | 0.450 | 0.310 |
| RAE | 0.614 | 0.764 | 0.854 | 0.703 | 0.312 | 0.433 | 0.561 | 0.396 |
| HypE | 0.662 | 0.800 | 0.844 | 0.737 | **0.403** | 0.531 | 0.652 | **0.489** |
| G-MPNN | 0.201 | 0.407 | 0.611 | 0.337 | - | - | - | - |
| **H$^2$GNN (Ours)** | 0.657 | **0.815** | **0.869** | **0.742** | 0.387 | **0.537** | **0.660** | 0.484 |

heterogeneous hypergraphs and it also compares favorably with the specialized embedding-based method HypE (Fatemi et al., 2020b), designed for link prediction tasks.

Table 4: Comparison Experiments: H$^2$GNN encodes the graph structure information and compares the experimental results of different decoders.

| Method | FB-AUTO | | | | JF17K | | | |
|---|---|---|---|---|---|---|---|---|
| | Hits@1 | Hit@3 | Hits@10 | MRR | Hits@1 | Hit@3 | Hits@10 | MRR |
| H$^2$GNN & HSimplE | **0.652** | **0.788** | **0.839** | **0.725** | **0.376** | **0.517** | **0.649** | **0.469** |
| HSimplE | 0.608 | 0.760 | 0.825 | 0.692 | 0.341 | 0.490 | 0.633 | 0.451 |
| H$^2$GNN & mTransH | **0.621** | **0.771** | **0.840** | **0.705** | **0.372** | **0.481** | **0.583** | **0.451** |
| mTransH | 0.602 | 0.754 | 0.806 | 0.688 | 0.370 | 0.475 | 0.581 | 0.444 |
| H$^2$GNN & m-DistMult | 0.657 | 0.815 | 0.869 | 0.742 | 0.387 | 0.537 | 0.660 | 0.484 |
| m-DistMult | 0.513 | 0.733 | 0.827 | 0.634 | 0.372 | 0.510 | 0.634 | 0.463 |

In addition, we conduct experiments to investigate the impact of different encoders and decoders on the link prediction task. As shown in Table 4, we fix H$^2$GNN as the encoder and combined it with three decoders: HSimplE, mTransH, and m-DistMult. The results demonstrate that the combination of H$^2$GNN & HSimplE and H$^2$GNN & m-DistMult significantly outperformed using HSimplE and m-DistMult alone on both datasets, indicating that the encoder-decoder synergy can better leverage the structural and semantic information of the knowledge graph.

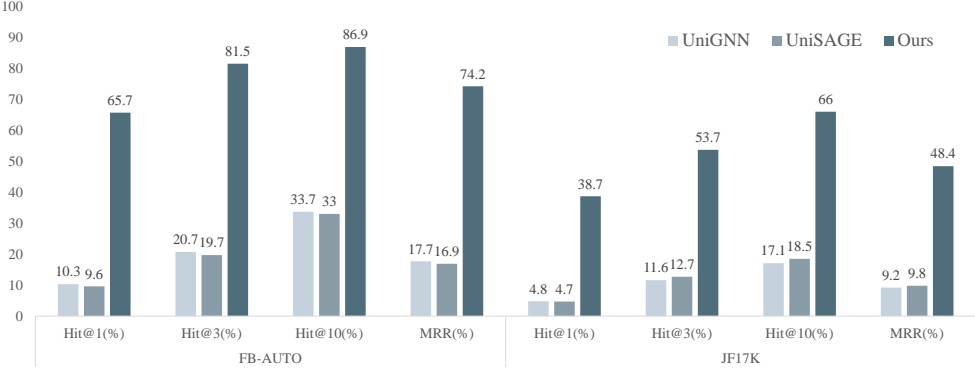

Figure 4: Comparison Experiments: Encoding the hypergraph structure information with different methods for the same m-DistMult decoding model.

Figure 4 compare the effects of different graph neural network encoders when paired with the m-DistMult decoder. The experimental results clearly demonstrated that H²GNN & m-DistMult significantly outperforms UniGNN & m-DistMult and UniSAGE & m-DistMult on both datasets. For instance, on the FB-AUTO dataset, the combination of H²GNN and m-DistMult achieves Hits@1 of 65.7% and MRR of 74.2%, while UniGNN and m-DistMult only reaches Hits@1 of 10.3% and MRR of 17.7%.

## 4.4 ABLATION STUDY

The ablation study examines the influence of the Hyperbolic Operation (HO) and Position Information (PI) modules in H²GNN on model performance, and the results are depicted in Figure 5. We conduct experiments using m-DistMult as the decoder in which we individually remove these two modules and compare them to the full H²GNN model. The experimental results clearly indicate that removing either module results in a significant deterioration in model performance. This suggests that both the HO and PI modules are effective and serve as complementary components.

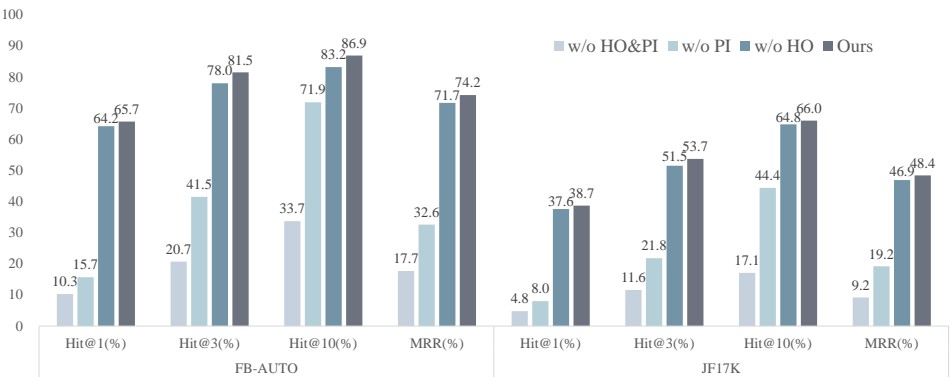

Figure 5: Module sensitivity on H²GNN for Hyperbolic Operation (HO) and Position Information (PI).

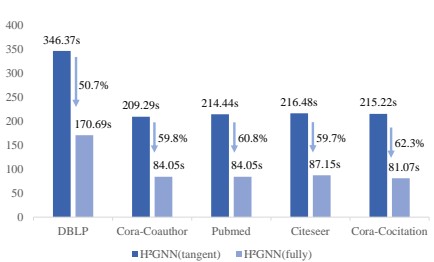

Figure 6: Comparison of the running time of H²GNN for full hyperbolic space and tangent space operations.

For further study, we compare the runtime performance of the H²GNN method when operating in different representation spaces: hyperbolic space and tangent space. The accuracy comparison is presented in the Appendix A.2. The operation in tangent space is a hybrid approach, where the features are transformed between hyperbolic space and tangent space by a series of hyperbolic and inverse hyperbolic mapping functions, and neural operations are performed in tangent space. As shown in Figure 6, the execution time in fully hyperbolic space is reduced by 50%-60% compared to the tangent space.

## 5 RELATED WORK

In this section, we review the representative (hyper)graph neural network techniques.

**Graph Neural Networks.** Research in graph neural networks serves as the foundational basis for GNN development. For instance, Graph Convolutional Networks (GCNs) (Kipf & Welling, 2017) leverage node degrees to normalize neighbor information. PPNP (Klicpera et al., 2019) tackles the over-smoothing problems in GNNs through skip-connections, and AdaGCN (Sun et al., 2021) integrates a traditional boosting method into GNNs.

Heterogeneous graph neural networks (Hu et al., 2020; Wang et al., 2019; 2023b; 2020; Zhang et al., 2019) have made significant strides in effectively addressing complex heterogeneity through the integration of message passing techniques. Notably, the Heterogeneous graph Propagation Network (HPN) (Ji et al., 2023) theoretically provides a theoretical analysis of the deep degradation problem and introduces a convolution layer to mitigate semantic ambiguity.

**Hyperbolic Graph Neural Networks.** Hyperbolic neural networks have demonstrated their ability to effectively model complex data and outperform high-dimensional Euclidean neural networks when using low-dimensional hyperbolic features (Dasgupta & Gupta, 2003; Giladi et al., 2012; Assouad, 1983). While existing hyperbolic networks, such as the hyperbolic graph convolutional neural network (Chami et al., 2019), hyperbolic graph neural network (Liu et al., 2019) and multi-relation knowledge graphs like $M^2$GNN (Wang et al., 2021b) and H2E (Wang et al., 2021a), encode features in hyperbolic space, they are not fully hyperbolic since most of their operations are formulated in the tangent space, which serves as a Euclidean subspace. In contrast, fully hyperbolic neural networks, such as FFHR (Shi et al., 2023) define operations that are entirely performed in the hyperbolic space, avoiding the complexities of space operations.

**Knowledge Hypergraph Neural Network.** Existing knowledge hypergraph modeling methods are derived from knowledge graph modeling methods, which can be primarily categorized into three groups: translational distance models, semantic matching models, and neural network-based models.

Translational distance models treat hyper-relations as distances between entities and formulate score functions based on these distances. For instance, models like m-transH (Wen et al., 2016) and RAE (Zhang et al., 2018) generalize the TransH model. They calculate a weighted sum of entity embeddings and produce a score indicating the relevance of the hyper-relation. Neural Network-Based Models, like NaLP (Guan et al., 2019) and NeuInfer (Rosso et al., 2020b), represent hyper-relations using main triples and attribute pairs. They calculate compatibility scores between the main triples and between the main triples and each attribute pair individually using neural networks. The final hyper-relation scores are determined based on these computations. Semantic Matching Models, such as HypE (Fatemi et al., 2020b) and GETD (Liu et al., 2020), assess the semantic correlation between entities and hyper-relations through matrix products. For instance, HypE builds upon SimplE (Rosso et al., 2020b) by incorporating convolution for entity embedding and employing multi-linear products for calculating plausibility scores.

## 6 CONCLUSION

In this paper, we represent knowledge facts as hypergraphs and introduce graph neural networks for knowledge hypergraphs and hyper-relations modeling. We propose Hyperbolic Hypergraph GNN, a method that directly encodes adjacent hyperedges and entity positions within knowledge hypergraphs. By considering both structural and positional information, we can accurately represent semantics. We also present the hierarchical structure of hyper-star message passing process in a fully hyperbolic space, which reduces distortion and boosts efficiency. Our $H^2$GNN encoder yields results comparable to the baselines for knowledge hypergraph link prediction and outperforms the state of the art for node classification and inductive learning on evolving hypergraphs tasks.

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

# A    MORE EXPERIMENTAL RESULTS

## A.1    INDUCTIVE LEARNING RESULTS

We use a corrupted hypergraph that randomly removes $40\%$ nodes as unseen data during training, following the approach (Arya et al., 2020). For training, We use $20\%$ of the nodes, reserving $40\%$ for the testing nodes that are seen during training. $H^2$GNN consistently outperforms other methods across benchmark datasets, achieving a remarkable accuracy of $89.7\%$ on DBLP, with a low standard deviation ranging from $0.1\%$ to $1.4\%$. This highlights the effectiveness and robustness of $H^2$GNN for inductive learning on evolving hypergraphs. $H^2$GNN can dynamically update the hypergraph structure and node features, leveraging high-order neighbor information to enhance node representation.

Table 5: The accuracy(%) results for inductive learning on evolving hypergraphs across Co-authorship and Co-citation datasets. We highlight the best and competitive results achieved by baselines and $H^2$GNN for each dataset.

| Method | DBLP | | Pubmed | | Citeseer | | Cora (Co-citation) | |
|---|---|---|---|---|---|---|---|---|
| | seen | unseen | seen | unseen | seen | unseen | seen | unseen |
| UniGIN | 89.4±0.1 | 83.2±0.2 | 84.5±0.3 | 83.1±0.4 | 69.1±1.1 | 68.8±1.7 | 71.6±2.0 | 68.7±2.1 |
| UniSAGE | 89.3±0.2 | 82.7±0.3 | 80.3±1.0 | 79.2±0.8 | 67.9±1.5 | 68.2±1.2 | 70.5±1.1 | 66.3±1.4 |
| UniGCN | 88.1±0.2 | 82.1±0.1 | 17.6±0.3 | 17.8±0.3 | 22.1±0.8 | 22.4±0.8 | 15.6±0.9 | 15.8±0.9 |
| UniGAT | 88.0±0.1 | 15.8±0.2 | 30.0±0.4 | 17.8±0.2 | 44.2±0.6 | 22.5±0.6 | 48.3±1.0 | 15.8±0.5 |
| **$H^2$GNN (Ours)** | **89.7±0.1** | **83.4±0.2** | **86.2±0.2** | **85.5±0.5** | **70.2±1.3** | **69.2±1.0** | **75.3±1.4** | **72.1±1.2** |

## A.2    COMPARISON OF NODE CLASSIFICATION ACCURACY IN $H^2$GNN ACROSS TWO SPACES

Table 6 illustrates that the fully hyperbolic space exhibits yields a modest improvement in accuracy performance. While the difference may not be substantial, it underscores the superiority of $H^2$GNN and the effectiveness of the hyper-star message passing process.

Table 6: Comparison of $H^2$GNN Model Accuracy (%) in two spaces: fully hyperbolic space vs. tangent space transformation, which the model encodes features in hyperbolic space but primarily conducts operations in the tangent space, a Euclidean subspace originating from the hyperbolic model.

| Method | Co-authorship Data | | Co-citation Data | | |
|---|---|---|---|---|---|
| | DBLP | Cora | Cora | Pubmed | Citeseer |
| $H^2$GNN (tangent) | 88.76±0.14 | 74.07±1.34 | 67.29±2.09 | 74.67±1.16 | 61.13±1.18 |
| $H^2$GNN (hyperbolic) | 89.75±0.20 | 74.97±1.20 | 69.43±1.54 | 74.89±1.23 | 62.52±1.48 |

