# OpenReview forum: "Fully Hyperbolic Representation Learning on Knowledge Hypergraph"
_ICLR.cc/2024/Conference — Submitted to ICLR 2024_

### Official Review · Reviewer_pg5m · 2023-10-28

**Soundness:** 2 fair
**Presentation:** 3 good
**Contribution:** 2 fair
**Rating:** 3
**Confidence:** 5

**Summary:**

This paper proposes a method called Hyperbolic Hypergraph GNN (H2GNN) for representation learning on knowledge hypergraphs. The authors argue that existing methods for knowledge hypergraph representation learning either transform hyperedges into binary relations or ignore the adjacencies between hyperedges, resulting in information loss and sub-optimal models. To address these issues, H2GNN uses a hyper-star message passing scheme to capture the semantics of hyper-relations, and it works in the fully hyperbolic space to reduce distortion and improve efficiency.

**Strengths:**

1) The paper proposes a novel method, H2GNN, for representation learning on knowledge hypergraphs. It addresses the limitations of existing methods by considering the adjacencies between hyperedges and explicitly incorporating entity positions.
2) H2GNN introduces the hyper-star message passing scheme, which is a novel approach to capture the semantics of hyper-relations.
3) The paper demonstrates the effectiveness of H2GNN through extensive experiments on node classification and link prediction tasks, comparing it with state-of-the-art baselines on various datasets.

**Weaknesses:**

1) For innovation, this paper lies mainly in the manipulation of star extensions and hyperbolic transformations of hypergraphs and is not very innovative.
2) The article lacks logic and does not reflect the necessity of combining the star extension and hyperbolic operation of the hypergraph. Hypergraphs themselves have a hierarchical structure, and following the logic in the article it would be possible to get low distortion results by learning hypergraphs directly on hyperbolic space.
3) Insufficient theoretical depth in experimental analyses. The good results of an experiment do not necessarily mean the quality of the article. It is important to start with the results and further compare and analyze them to explore the reasons why the method is SOTA.

**Questions:**

1) please elaborate on the reasons that hyperbolic space outperforms Euclidean space, such as how the distortion is reduced. The argument of the article is that the graph structure after the star extension of the hypergraph structure has a hierarchical structure and therefore has less distortion in the hyperbolic space. But the hypergraph structure and the graph structure after the star extension are bijective equivalences, so why must one need to combine the hyperbolic operation with the star extension? The writer needs to sort out the logic of the article again.
2) Is it possible to visualize star-expanded hypergraphs on hyperbolic space? This would enhance the persuasiveness of the article.
3) Can you further analyze in your experiments why your method can perform better compared to other hypergraphic learning methods?

---

### Official Review · Reviewer_p6a6 · 2023-11-03

**Soundness:** 2 fair
**Presentation:** 1 poor
**Contribution:** 2 fair
**Rating:** 3
**Confidence:** 4

**Summary:**

In this work the authors propose a hypergraph-specific message passing scheme intended to capture higher-order relations in knowledge hypergraphs. The main novel contribution of the authors is their Hyper-star Message Passing algorithm, which describes how to update hyperedge and node representations in the hyperboloid $\mathbb H_k^n$. Given existing node features $\\{\mathbf x_i\\} \subseteq \mathbb H_k^n$ and a hyperedge $e = (r, x_1, \ldots, x_m)$, the authors define the representation of this hyperedge as the centroid (in the hyperboloid) of $\\{x_i\\}_{i=1}^m$.

To get an updated representation of node $x_i$, they first consider $\mathcal E_i$, the set of all hyperedges containing $x_i$. From each $e \in \mathcal E_i$, they create $\mathbf h_p \in \mathbb R^d$ which is a relation-type and position-specific embedding based on the position of $x_i$ in $e$. They then update the node embedding for $x_i$ by calculating the centroid of $\mathbf x_i$ and $\\{\mathbf h_e - \mathbf h_p\\}_{e\in\mathcal E_i}$ .

The authors evaluate this message passing approach on two tasks - node classification, and link prediction - where they observe improvements over a number of baselines. They also perform a comparative study evaluating different decoders for link prediction as well as different encoders, as well as an ablation of the model which suggests both the hyperbolic aspects of the model and position information of the model are beneficial for link prediction.

**Strengths:**

The proposed hypergraph-specific message passing scheme is original, and intuitively seems to be a reasonable approach to capturing information in the hypergraph setting. The formulation of their model allows for a flexible encoder/decoder architecture which the authors highlight in one of their comparative evaluations.

**Weaknesses:**

Unfortunately, I was not able to fully understand the author's proposed model. For example, in section 3.1 the authors talk about implementing a "matrix function to perform linear transformations in hyperbolic space", but it is not clear to me when this was used. In looking at the diagram of the model in Figure 3, it looks like a hypergraph is passed as input to the linear transformation, which is then passed to the message passing component, but this does not make sense to me. I would love to see a clear step-by-step explanation of the model, taking a hypergraph as input and outputting node embeddings.

The empirical evidence is also somewhat mixed. The authors claim their model "significantly outperforms other methods on all datasets", but actually it is relatively close (and almost always within one standard deviation) to UniSAGE. The link prediction results are somewhat similar - while the author's proposed model performs well, HypE is a close contender (and outperforms in a few cases). Link prediction is also a task fraught with evaluation difficulties, with test set bias often explaining minor differences in performance.

**Questions:**

1. Could you provide a clear step-by-step explanation of how your model starts from a hypergraph and creates node embeddings?
2. I was surprised that the position-specific embedding modifies the representation of a hyperredge in the aggregation for creating a new node representation as opposed to modifying the representation of the node  in the aggregation for creating a hyperedge representation. Is there some benefit to your proposed approach compared to the one I describe here?
3. The notation for $\mathbf h_p$ is confusing. First, it is stated that $\mathbf h_p \in \mathbb R^d$, but then it is subtracted from a vector in $\mathbb H$. Secondly, the dependencies are not clear from the notation. From the text, it is clear that $\mathbf h_p$ is dependent on both a relation and a position, but in equation (4) this is left to determine from context. I assume that in equation (4), $\mathbf h_p$ is different for each $e$ and depends on the relation in $e$ as well as the position of $x_i$ in $e$, but the current notation really doesn't make this clear.
4. In section 3.3 the loss functions are presented, but it is not explained how the encodings from the message passing algorithm are converted to conditional probabilities equation (6) or confidence scores $g(x)$ in equation (7). I assume an MLP in the first case, and I see that there is a later discussion about decoders for link prediction, but details here are lacking.
5. To my understanding, m-DistMult is not particularly well-aligned to extract information from H^2GNN, yet it is shown in Figure 4 that it performs far worse when paired with UniGNN or UniSAGE. Were these encoders trained with m-DistMult as the decoder, or are these the result of simply using m-DistMult on top of frozen encoders? What is the explanation for the incredibly poor performance of UniGNN and UniSAGE when using m-DistMult?
6. In section 4.4 there is an ablation of Hyperbolic Operation (HO) and Position Information (PI) modules, which is the first time these terms are mentioned in the text. What are these modules? (This goes back to my primary question related to a clear step-by-step structure of the model.)

**Typos / Suggestions**

* Page 2: Additional criteria are needed to complete this statement: "The goal of representation learning is to obtain embedded representations for each node x ∈ V and relation type r ∈ R in the knowledge hypergraphs." As it stands, this could be satisfied by randomly assigning a vector to each node. The goal here should include some additional objective, eg. "... such that they can be used to reconstruct the original graph".
* Page 3: "In Figure 2, when we embed the tree structure into Euclidean space, the distance between the yellow and pink nodes on the tree is 8 nodes apart, but in reality, these tree-like structures are very close in real networks." I do not understand the latter part of this sentence. As I have understood it, the problem is typically the reverse - that embedding a tree in Euclidean space makes leaves from different paths closer than they should be, which is exactly as depicted in Figure 2 and also concordant with the sentence which follows this one in your paper.
* Page 4-5: The notation $\\{x_j\\}_{j\in e}$ doesn't make sense to me, because the integer $j$ is not in $e$. At best, I could see something like $\\{x_j \mid x_j \in e\\}$, or maybe just using $\\{x_1, \ldots, x_m\\}$ as the argument to the centroid function and putting "for $e = (r, x_1, \ldots, x_m)$" to the right or in the text above the block equation.
* Page 5: The squared Lorentzian norm is mentioned but has not been defined, it would be good to include the definition for completeness.
* Page 6: I would advise against claiming the model "significantly outperforms" the baselines when, in fact, many of the results are within 1 standard deviation of the UniSAGE baseline. Also, the accuracy range of 89.75% to 74.89% quoted in the text is incorrect based on the numbers in Table 2.

---

### Official Review · Reviewer_wicp · 2023-11-10

**Soundness:** 2 fair
**Presentation:** 2 fair
**Contribution:** 1 poor
**Rating:** 3
**Confidence:** 5

**Summary:**

This paper developed the Hyperbolic Hypergraph GNN (H2GNN) for modeling complex relationships in knowledge hypergraphs using hyperedges. This method utilizes hyper-star message passing for a non-lossy hierarchical representation of hyperedges, embedding them in hyperbolic space for reduced distortion and improved efficiency. H2GNN outperforms well in experiments, but it does not compare with the strong baselines.

**Strengths:**

The paper is well-organized.

The paper provides a comprehensive literature review.

**Weaknesses:**

The paper is not organized clearly, which is not easy to understand. For instance, there is a lack of details for hyperbolic space explanation.

There lacks a comparison between the strong baselines, such as [1] [2], which makes the performance not comparable.

[1] Searching to sparsify tensor decomposition for n-ary relational data

[2] Role-aware modeling for n-ary relational knowledge bases.

**Questions:**

Please refer to the Weakness.

---

### Meta-Review · Area_Chair_77yc · 2023-11-26

**Metareview:**

The paper proposes a new representation learning method and a message passing algorithm for hypergraphs.


Weakness: The method is close to the well-known star extension and lacks of clarity to explain the experimental results, in particular there is no theoretical insight and no fair comparison with competing methods.

The authors did not reply to questions from the reviewers. I consider that they adhere to the comments that have been given.

**Justification For Why Not Higher Score:**

The authors did not reply to questions from the reviewers. I consider that they adhere to the comments that have been given.

**Justification For Why Not Lower Score:**

N/A

---

### Decision · Program_Chairs · 2024-01-16

Reject